# Immune Tolerance of Embryo Implantation and Pregnancy: The Role of Human Decidual Stromal Cell- and Embryonic-Derived Extracellular Vesicles

**DOI:** 10.3390/ijms232113382

**Published:** 2022-11-02

**Authors:** Hsien-Ming Wu, Liang-Hsuan Chen, Le-Tien Hsu, Chyong-Huey Lai

**Affiliations:** 1Department of Obstetrics and Gynecology, Linkou Medical Center, Chang Gung Memorial Hospital, Chang Gung University College of Medicine, Taoyuan 333, Taiwan; 2Gynecologic Cancer Research Center, Linkou Medical Center, Chang Gung Memorial Hospital, Taoyuan 333, Taiwan

**Keywords:** decidual stromal cell, endometrium, extracellular vesicle, immunomodulation, embryo implantation, pregnancy

## Abstract

Embryo–endometrial communication plays a critical role in embryo implantation and the establishment of a successful pregnancy. Successful pregnancy outcomes involve maternal immune modulation during embryo implantation. The endometrium is usually primed and immunomodulated by steroid hormones and embryo signals for subsequent embryo implantation and the maintenance of pregnancy. The roles of extracellular vesicles (EVs) and microRNAs for the embryo–maternal interactions have been elucidated recently. New evidence shows that endometrial EVs and trophectoderm-originated EV cargo, including microRNAs, proteins, and lipids in the physiological microenvironment, regulate maternal immunomodulation for embryo implantation and subsequent pregnancy. On the other hand, trophoblast-derived EVs also control the cross-communication between the trophoblasts and immune cells. The exploration of EV functions and mechanisms in the processes of embryo implantation and pregnancy will shed light on a practical tool for the diagnostic or therapeutic approaches to reproductive medicine and infertility.

## 1. Introduction

The embryo–endometrial communication has a critical role in the successful embryo implantation and pregnancy [1]. Maternal immune modulation during embryo implantation and subsequent pregnancy provides a tolerant microenvironment to keep the fetus from rejection [2]. Since the first human IVF baby was born in 1978, pregnancy and live birth rates following assisted reproductive technologies have improved significantly. Even so, there is still a gap in achieving higher pregnancy and live birth rates, attributed to unresolved problems in embryo quality, endometrium receptivity, and embryo–endometrium interaction [3,4]. Usually, a steroid hormone primes the endometrium for the receptive phase; after that, it is further immunomodulated by embryo signals for subsequent embryo implantation and maintenance of pregnancy [5]. Therefore, successful embryo implantation and the ensuing pregnancy depend on the coordinated interaction between the endometrium and embryos. The classical embryo–endometrial interactions through dynamic intracellular and secreted protein alterations in the stage of embryo implantation are performed to promote the successful maintenance of pregnancy [6]. The role of extracellular vesicles (EVs) and microRNAs in the decidualization for the embryo–maternal interactions of decidual endometrial stromal cells has been clarified in the implantation of the embryos [7]. EVs are cell-originated membranous vesicles that deliver bioactive molecules from cells to cells [8]. The cargo of EVs mediates cell–cell communication by delivering numerous factors, including DNA, microRNAs, proteins, and lipids [9]. Innovative insights and approaches have explored the broad features of EVs, including ligands, surface receptors, and cargo from the mother cells. Moreover, the lipid bilayer membranes of EVs contain relatively high levels of ceramide, cholesterol, and sphingomyelin, causing EVs to be steady in extracellular spaces [10]. The biochemical and molecular characteristics of EV cargo from cells of the female reproductive tissue, placenta, and embryo have been recently demonstrated [7,11,12]. In this review, we will highlight the roles of human decidual stromal cell-associated EVs in the immune tolerance of embryo implantation and pregnancy.

## 2. Decidualization of the Human Endometrium

The endometrium experiences regeneration, differentiation, and shedding during the human menstrual cycle. Usually, the endometrium in the menstrual period undergoes three main stages: the proliferative phase, the secretory phase, and the menstrual phase. Two ovarian steroid hormones, estrogen (E2) and progesterone (P4), regulate the endometrial cycle. E2 and P4 functionally and morphologically control the endometrial epithelial, stromal, and immune cells [13]. After the E2-primed proliferative phase, the proliferated endometrial epithelial and stromal cells differentiate into decidual cells in response to increasing P4 in the secretory phase [14]. The corpus luteum-synthesized and released P4 is required for promoting embryo implantation and the maintenance of placentation [15].

### 2.1. Morphological Differentiation in the Human Endometrium

During embryo implantation and pregnancy, the endometrium experiences decidualization in which the endometrial epithelium, stroma, and vascular structures are transformed into decidua [16,17]. The decidualization results from the elevated levels of E2 and P4 and results in a progressive change in tissue remodeling, cellular functions, and gene expression until the completion of placentation during pregnancy. During the process of decidualization, vascular endothelium, immune cells, glandular epithelium, and endometrial stromal cells react to the integration of multiple factors and mediators. After ovulation, the human endometrial stromal cells transform from fibroblast-like cells to epithelium-like cells with large pale nuclei, cytoplasmic expansion, and globular shapes in the secretory phase [18]. This transforming process features intricate cytoskeletal rearrangements [19]. The complex cytoskeletal rearrangements involve the activation of a myosin light chain and condensed F-actin, promoting intracellular remodeling [20]. P4-transformed epithelium-like decidual stromal cells produce decidual proteins, such as prolactin (PRL), insulin-like growth factor-binding protein-1 (IGFBP-1), and leukemia inhibitory factor (LIF) [21,22,23].

### 2.2. Functional Differentiation in Human Endometrium

Progesterone receptor (PR) expression plays a role in the signaling pathways supporting endometrial homeostasis for embryo implantation and pregnancy. PRs are nuclear receptors including two isoforms (PR-A and PR-B) with disparate functions [24]. PRs appear throughout pregnancy [25], indicating that the expression level of the PRs in the human decidua is the consequence of an integrated E2 and P4 regulation and autoregulation of its promoter region [26,27]. Ligand-binding PRs are enrolled to P4 response elements in the promoters of target genes and regulates their transcription [28]. PR pathways with accumulations of cAMP promote the expression of decidual transcriptional regulators, epigenetic modifications, coordination of signal transductions, and post-transcriptional moderation [17]. The physiological role of the dynamic expression of PRs in the endometrium as decidualization initiation and ongoing pregnancy could denote the intrinsic mechanism for a functionally dynamic progesterone change [26]. The initiation of the decidualization induces decidual stromal cells to produce angiogenic factors, growth factors, cytokines, and chemokines related to blastocyst implantation.

Consequently, decidual stromal cells release numerous proteins, such as IGFBP-1 and PRL. IGFBP-1 and PRL induce trophoblast invasion and proliferation through the integrins and PRL receptors [29,30]. Through PRL, IGFBP-1, fibronectin, and PR response element signaling, the integrated regulation between the steroid hormones and their receptors delivers important messages that govern decidual transformation and endometrial immune homeostasis for embryo implantation and pregnancy [31,32]. LIF and interleukin (IL)-11 upregulate the expression of adhesion molecules and the adherence of endometrial epithelial cells to collagen IV and fibronectin [33]. LIF activates the adhesion molecules in trophoblasts and increases the attachment of trophoblasts to laminin and fibronectin (elements of the extracellular matrix); meanwhile, IL-11 enhances the adherence of endometrial epithelial cells to trophoblasts [33,34,35]. These LIF- and IL-11-induced effects promote the adhesion and invasion of the blastocyst and maintain the subsequent placentation [34,36].

## 3. Immune Modulatory Properties of Decidual Stromal Cells

The homeostasis between active immunity and tolerance at the maternal–fetal surface between uterus and embryo is significantly important. Therefore, effective immunity must be supported to protect the mother from external organisms, and tolerance should be provided toward embryos. Decidual stromal cells may play an essential role throughout the pregnancy, and to prevent an allogeneic response against the embryo and fetus, it is crucial to establish a tolerant microenvironment required to inhibit maternal immune response and keep the fetus from rejection [37,38]. Decidual stromal cells behave as the major cellular component of the human decidua and derive from a fibroblast-like stromal cell undergoing differentiation and proliferation [39]. Decidual stromal cells can manifest stem cell properties, including multilineage differentiation into three embryonic layers [40]. Furthermore, decidual stromal cells have immune modulatory properties on cells of innate and adaptive immunity in humans (Table 1). In humans, decidual stromal cells have been demonstrated to inhibit T cell function and to promote Tregs through the activation of indoleamine-2,3-dioxygenase (IDO), prostaglandin E2, programmed death ligand (PD-L)1, and interferon-gamma (IFN-γ) [41]. Meanwhile, peripheral blood (PB) CD14+ monocytes are differentiated into tolerogenic dendritic cells (DCs), expressing low CD83 and CD86 and releasing IL-10 in the presence of decidual stromal cells [42]. These decidual stromal cells failed to stimulate T cell proliferation while inducing T cell differentiation into IL-4+ Th2 rather than IFN-γ+ Th1 cells. According to the immunomodulatory effect, the application of decidual stromal cells has been shown to be effective in treating immune disorders [43,44].

### 3.1. Uterine Natural Killer (uNK) Cells

NK cells are classified as innate immunity and are categorized into cytotoxic NK cells and noncytotoxic NK cells [45]. uNK cells act as neoangiogenesis, tissue remodeling, and regulation of trophoblast invasion during the first trimester of pregnancy [46]. uNK cells are noncytotoxic and constitute about 70% of all leukocytes in the uterus. The recruitment mechanisms of uNK cells have been demonstrated such that NK cells migrate from the peripheral blood vessels to the uterine decidua; furthermore, uNK cells might result from the differentiation of permanent tissue CD34+ hematopoietic precursor cells [47,48]. uNK cells are highly reactive to environmental stimulation with notable functional characters in early pregnancy decidua [49,50]. uNK cells deliver the balanced cell function between activating and inhibitory stimulation, indicating that uNK cells express an intricate mechanism of activating and inhibitory receptors that serve as reactions to specific ligands in target cells or responses to other cells of the host organism to establish self-tolerance and acquire functional capability [51]. CD56bright and CD16-uNK cells function as the setting and maintenance of early pregnancy through the motivation of tissue remodeling, angiogenesis, and trophoblast invasion in the decidua [52,53]. Recent reports demonstrate new insights into the crucial role of innate immune cells in avoiding the senescence in the human endometrium occurring in early pregnancy to prevent recurrent pregnancy loss through human uNK cells in the decidua [54,55]. Concerning the mechanisms of the uNK cell action, exploring the major histocompatibility complex (MHC) class I-specific inhibitory and activating uNK receptors, identifying ligands provided the way to understand the signaling pathway [56]. The human leukocyte antigen (HLA) profile of fetal extravillous trophoblast cells (EVTs) is unique with polymorphic HLA-C class I molecules, HLA-E and HLA-G, but without class I HLA-A and HLA-B or class II molecules [57]. Moreover, EVT HLA ligands interact with uNK cell receptors, such as killer cell immunoglobulin-like receptors (KIRs) [57,58]. The tissue remodeling, angiogenesis, and trophoblast invasion, with subsequent impacts on placentation and the maintenance of pregnancy, may be promoted by the allorecognition between paternal HLA-C and maternal KIRs [59,60]. Some reproductive outcomes, such as embryo implantation failure, recurrent pregnancy loss, fetal growth restriction, and preeclampsia, have a similar pathogenesis of abnormal angiogenesis and tissue remodeling, associated with maternal KIRs and fetal HLA-C genotypes [61,62]. EVTs release progesterone and profilin-1, which function as the inducer of endometrial stromal cell (ESC) decidualization through the downregulation of arachidonate 5-lipooxygenase (ALOX5) in ESCs [63].

### 3.2. Cytokines

Functional embryo–maternal interactions occur during embryo implantation and placentation, contributing to a successful pregnancy [64]. Embryo implantation is a vital step in the complex embryo–maternal interactions, indicating that a well-ready and receptive endometrium recognizes and tolerates a good-quality embryo [65]. Cytokines released from the endometrium regulate these complex embryo–maternal interactions throughout pregnancy as the immunoregulatory roles in decidualization and placentation [66]. In order to prime a receptive endometrium, several immune cells, including uNK cells and macrophages, are initiated to release growth factors and proinflammatory cytokines modulating receptive decidualization and maternal tolerance to embryo invasion [67,68,69]. All through the pregnancy, cytokines establish complex immunological stability between inflammatory and anti-inflammatory reactions essential for balancing embryo–maternal interactions [70]. During embryo implantation and pregnancy, the coordination of different immune cells and cytokines in the endometrial immune environment plays a crucial role [71]. It has been demonstrated that uNK cells, macrophages, and T cells are most of the leukocyte in the decidua throughout embryo implantation and pregnancy [50,67,72]. Most immune cells in the early proliferative endometrium are T cells and are classified into helper CD4+ T cells and cytotoxic CD8+ T cells [73]. Various cytokines, which regulate and modulate the immune reactions in the maternal–fetal interface, are secreted by both Th1 and Th2 cells of the CD4+ T cell subtypes [74]. These cytokines can be categorized as proinflammatory Th1-type and anti-inflammatory Th2-type based on their style of paracrine regulation [75]. The Th1-type cells release proinflammatory cytokines, mainly interleukins (IL)-1, 2, 12, 15, 18, and IFN-γ), and tumor necrosis factor-alpha (TNF-α). The Th2-type cells release anti-inflammatory cytokines, mainly granulocyte-macrophage colony-stimulating factor (GM-CSF), IL-4, 5, 6, 10, and 13 [67,72].

Furthermore, according to the type of cytokines secreted from the macrophages, the cells are divided into proinflammatory and anti-inflammatory macrophages, which involve trophoblast invasion and vascular and tissue remodeling [76]. Various cytokines synchronize and balance the maternal–fetal immune system modulation, involving embryo implantation and pregnancy maintenance. This unique mechanism is featured by the dynamic production of different cytokines, which means the dynamic changes of the Th1/Th2 ratio during the different phases of embryo implantation and pregnancy [77,78]. An inflammatory environment is organized in the decidua through the maternal–fetal immune system modulation to induce adequate embryo implantation and placentation. Meanwhile, the release of proinflammatory Th1 cytokines during this stage is essential for embryo implantation and maintenance of pregnancy since this inflammatory environment induces extravillous trophoblast invasion, vascular remodeling, and maternal–fetal immune tolerance [79]. Consequently, a shift of the Th1/Th2 immune response throughout the peri-implantation period promotes embryo implantation and placentation [78]. Subsequently, to maintain the pregnancy, a predominating anti-inflammatory Th2 condition establishes normal fetal growth and development, as well as sheltering both the mother and the fetus from potential inflammation [78]. During the third phase of pregnancy, a renascent inflammation that coordinates with a proinflammatory environment initiates labor associated with inducing uterine contraction, infant delivery, and placenta expulsion [79,80]. Taken together, the dynamic changes of the Th1/Th2 ratio during the different pregnancy phases achieve a maximum in the proliferative and primed endometrium for embryo implantation; after that, the ratio decreases during the maintenance of pregnancy and eventually increases again during the initiation of labor course.

IL-15 has an essential role in maternal–fetal interactions through binding the receptors of uNK cells [81]. Many studies have shown that IL15 is significantly expressed during the secretory phase in the human endometrium through quantitative polymerase chain reaction (PCR) and histological analysis [18,82]. Furthermore, during the progestin-induced decidualization, IL-15 is highly expressed in cultured human ESCs [83]; meanwhile, IL15 is suggested as an important marker of the endometrial window of embryo implantation in the endometrial receptivity array [84]. Recently, it was also demonstrated that the IL-15 and IL-15RB/G receptor interaction is part of the crucial mechanisms between decidual stromal cells and uNK cells by single-cell analysis during first trimester human pregnancy [57]. Consequently, these reports demonstrate that IL-15 secreted from ESCs plays an important role in the modulation of the differentiation and function of uNK cells in the human endometrium during embryo implantation and placentation. Taken together, malfunctions in the endometrial immune system regulation, mainly resulting from an abnormal expression of Th1/Th2 cytokines or IL-15, are the predisposing factors for complicated pregnancy outcomes, which include embryo implantation failure, miscarriage, preterm labor, preeclampsia, and intrauterine growth restriction of the fetus [78,85,86] (Figure 1).

### 3.3. T Cells

Various T cells are present in the uterus and the decidua. T cells expressing different T cell receptors are defined as distinct subtypes, which include CD4+ T cells, CD4+ T regulatory cells, and CD8+ T cells [87,88].

CD4+ T cells were observed in the peripheral blood with a higher activated memory type in pregnant women compared to those in nulliparous women and maintained a high level one year postpartum [89,90]. Many studies in mice and humans demonstrated that these pregnancy-promoting memory CD4+ T cells could improve reproductive outcomes observed in multiparous pregnancies [91]. CD4+ T cells directly recognize MHCII-bound antigens and indirectly recognize APC-presented antigens in the decidua. CD4+ T cells reveal higher levels of CXCR3 (chemokine receptor) in the decidua to recruit effector memory T cells into the decidua [92]. CD4+ T cells can secrete cytokines in the uterine immune system, in which Th1 and Th17 cells release IFN-γ and IL-17 to promote trophoblast invasion and vascular and tissue remodeling of the uterine spiral arteries and the decidua [93,94]. These dynamic changes and balance of the cytokine secretion also contribute to inflammation and complicated pregnancy outcomes [95].

CD4+ T regulatory cells (Treg cells), which were formerly defined as suppressor T cells, with the co-expression of the transcriptional regulator forkhead box protein P3 (FOXP3), contribute to modulating the immune system, which includes preserving tolerance to self-antigens, avoiding autoimmune disease, and dynamically enhancing the decidua during embryo implantation and early pregnancy [96], but they seem to be unnecessary for the maintenance of the third trimester pregnancy in humans [97,98,99]. The recruitment of Treg cells in the decidua and the increase in maternal blood are observed from the first trimester of human pregnancy; however, an inadequate accumulation of Treg cells with impaired function may be associated with poor reproductive outcomes, such as implantation failure, miscarriage, and preeclampsia [100,101,102]. There are reports that the number of effector Treg cells in the decidua is increased in the third trimester of pregnancy compared to the first trimester of pregnancy, and more Treg cells in the decidua are present in pregnancies carrying an HLA-C-mismatched child compared to HLA-C-matched pregnancies [103]. Treg cells in the human decidua can reduce CD4+ T cell growth and cytokine secretion; meanwhile, through the release of IFN-γ and IL-17, Treg cells promote decidual vascular and tissue remodeling [92,104].

CD8+ T cells are dominant lymphocytes in the decidua during the second and third trimesters of pregnancy [57,88]. Notably, during the third trimester of pregnancy, CD8+ T cell receptors with an effector memory phenotype can identify fetal antigens offered by trophoblast HLA-C molecules similar to minor histocompatibility antigens [92,93,105]. In a successfully advanced pregnancy, the allogenic trophoblast is inadequate to actuate potential CD8+ T cell cytotoxicity; however, in the decidua, maternal CD8+ T cells are designated for HLA-A or -B and may cross-react with viral peptides bound to trophoblast HLA-C and drive the CD8+ T cell cytotoxicity [106]. In human pregnancy, CD8+ T cells reveal regulatory or inhibitory roles on effector CD8+ T cell reactions, which include part of the CD8+ T cells co-expressing the immune checkpoint molecules PD1 and TIM3 or negative for the costimulatory molecule CD28 [107,108]. Furthermore, HMOX1 was demonstrated in the placenta [109], which needs to be interpreted as the association between HMOX1 and CD8+ cells in immune regulation and maintenance during pregnancy.

### 3.4. Macrophages

Macrophages share several features present in most human tissue and function as an important role in tissue homeostasis, including immunity and tissue remodeling. In the uterine endometrium, macrophages are present to protect the embryo by expressing half allogenic antigens from the maternal immune reaction, indicating that macrophages in the decidua contribute to pathogen clearance, trophoblast invasion, and tissue and vascular remodeling for embryo implantation [110]. Therefore, the malfunction of macrophages results in abnormal reproductive outcomes, including implantation failure, miscarriage, preeclampsia, and other obstetrical complications [67,111,112,113,114]. A dynamic equilibrium of macrophage polarization is defined as emerging the phenotypic and functional features for internal and external stimulation. Macrophages were traditionally categorized into two phenotypes: classically activated macrophages (M1) and alternatively activated (M2) macrophages; however, the current report demonstrates that the plasticity of macrophages replies on microenvironmental stimulation with dynamic and continuous behavior in different tissues and organs [115,116]. In the decidua of embryo implantation and early pregnancy, macrophages may mainly contribute to immune regulation with tissue and vascular remodeling required for maintaining pregnancy by eliminating cell debris and apoptotic cells [57,117,118]. The cytokine profiles govern the function and phenotype of the macrophages’ response to microenvironmental stimulation. Usually, Th1 cytokines induce macrophages to M1 macrophages with highly inflammatory and phagocytic abilities; on the other hand, Th2 cytokines govern macrophages toward M2 macrophages with a different phenotype and immune regulatory functions [119]. There is acknowledgement that M1 macrophages are present to induce embryo attachment to the decidua during the peri-implantation period [118]. The macrophage profile shifts to an anti-inflammatory M2 macrophage-dominant environment to maintain the embryo implantation and placentation of early pregnancy [120]. The dysregulated transformation of the decidua M1–M2 macrophages results in poor reproductive outcomes, which include embryo implantation failure, miscarriage, preterm labor, and preeclampsia [96,121]. Furthermore, the highly flexible shift of M1–M2 macrophages may reveal more comprehensive functional plasticity [116]. Recently, an alternative classification was proposed according to the surface expression of molecules, the location, and gene expression profiles, such as CD11clow decidual macrophages [118]. Several factors, such as cytokines, hormones, and growth factors in the decidua, contribute to macrophage polarization; however, the underlying controlling mechanisms remain to be elucidated [122,123,124,125]. ALOX5 in macrophages is downregulated by Profilin-1, suggesting that Profilin-1 probably has a role in regulating cytokine secretion and immune tolerance [126]. Meanwhile, maternal macrophages are also observed in the trophoblast-originated placentas, probably resulting from moving maternal macrophages into the developing placenta in response to immune regulation. However, this phenomenon still needs to be probed [57].

### 3.5. Other Immune Cells

Dendritic cells (DCs) contribute to the connection between innate and adaptive immune reactions through the functions of DCs, such as presenting, capturing, and processing antigens, which is mainly accomplished by three DC subgroups: plasmacytoid DC and myeloid/conventional DC1 (cDC1) and 2 (cDC2) [127]. Conventional DCs (cDC1) are dominant compared to cDC2 in the human decidua. [57,117], in which the abundance of cDC1 may play a role in the increased frequencies of CD8+ dT cells, as cDC1 can crosspresent exogenous antigens, such as fetal antigens, bound to HLA-I molecules to actuate CD8+ T cells [57]. Meanwhile, in the absence of infections, the coexpression of PD1 in decidual cDC1 may abolish local CD8+dT cell activation [57]. The decidual DC phenotype may result from the reactions with extravillous trophoblast cells in the uterus of fertile women associated with Treg cells [117,128].

It is currently acknowledged that more B cells with increased levels of IgM, IgA, IgG3, and IgG4 are present in the peripheral blood of pregnant women compared to nonpregnant women [129]. B cells in the human decidua are a minority of immune cells, and they mildly increase with slight phenotypic and functional alterations throughout pregnancy by pregnancy hormones [88,130].

Higher markers for activated memory B cell phenotypes are observed in the decidua compared to the peripheral B cells at term labor, in which IL-10 production features the B cells with a regulatory phenotype [130]. The progesterone activity and IL-33 signaling drive the phenotype and function of B cells in physiological pregnancies [130]. These interactions regulate B cells to produce PIBF1, which may inhibit dNK cell activity, neutrophil infiltration and activation, and the production of proinflammatory mediators [131]. The role of B cells in pregnancy still needs to be extensively investigated.

**Table 1 ijms-23-13382-t001:** The roles of immune cells and cytokines in human decidua.

Immune Cells and Cytokines		Effect and Mechanism	Present in Uterus *	Role in Human Decidua
Cytokines [74,75,76]		Immunoregulatory	++	Modulating receptive decidualization, maternal tolerance to embryo invasion
Uterine natural killer cells [46,47,48]		Cytokine secretion, cytotoxicity	+++	Angiogenesis, tissue remodeling, regulation of trophoblast invasion
T cells	CD 4+ T cells [92,93,94,95]	Cytokine secretion, recruit effector memory T cells	++	Promote trophoblast invasion, vascular and tissue remodeling
	CD 4+ Treg cells [96]	Immune regulation	+	Promote decidual vascular and tissue remodeling
	CD 8+ T cells [107,108]	Cytotoxicity, cytokine secretion	++	Immune regulation
Macrophages [110]		Phagocytosis, antigen-presenting cells (APCs), cytokine secretion	+++	Pathogen clearance, trophoblast invasion, tissue and vascular remodeling
Other immune cells	Dendritic cell (DCs) [117]	Antigen-presenting cells (APCs)	+	T cell induction, immune tolerance
	B cells [129,130]	Antibody production, cytokine secretion	+	Immune regulation

* + (weak), ++ (moderate), and +++ (strong); this proves that some populations are more abundant than others, but no quantitative data are provided.

## 4. Identification of Decidual Stromal Cell-Associated Extracellular Vesicles

Extracellular vesicles (EVs) are in body fluids and tissues, which include blood, urine, and extracellular exudates. In the human decidua, EVs contain microRNAs, lipids, proteins, and glycosphingolipids. MicroRNAs are small noncoding RNAs that play critical regulatory roles in regulating gene expression and immune environment in decidual stromal cells for placentation and maintenance of pregnancy (Figure 2). Recent studies have demonstrated that several potential biomarkers for diagnosing pregnancy disorders result in the formation of proinflammatory environments and endothelial cell dysfunction in the decidua and placenta, which is associated with the level of maternal EVs [132]. The identification and characterization of EVs still need to be determined. Analyzing EV elements is challenging, depending on the extraction methods and the type of original cells [133,134]. Flow cytometry and nanoparticle tracking analysis (NTA) can identify the size and level of EVs; additionally, quantitative analysis of CD63, a typical EV marker, can further help recognize EVs [7,135]. Engineered EVs labeled with fluorescent dyes to the delivery system of the EVs within the maternal–fetal interface can elucidate tissue-derived EVs more sharply and directly [136]. There is evidence for the endometrial stromal cell-associated EVs, which have been purified from cultured endometrial stromal cells and decidual stromal cells [137,138,139]. These isolated EVs are about 100 nm in size and represent CD63 and TSG101 markers [137]. The components of endometrial stromal cell-derived EVs, such as tetraspanin-6 (TSPAN6), disintegrin, and metalloproteinase domain-containing protein 10 (ADAM10) are unique in response to hypoxic conditions as compared to normoxic conditions [137]. The characteristics of isolated EVs from endometrial stromal cells are demonstrated using nanoparticle tracking analysis (NTA) and electron microscopy (TEM). Moreover, positive EV protein markers CD63, TSG101, CD9, and CD81 are expressed by TEM and immunogold-EM analysis [7,140,141].

## 5. The Content of Extracellular Vesicles

Due to the cellular origin and pathophysiological condition, EVs deliver many different bioactive molecules, including DNA, RNA (mRNA and microRNA), protein, and lipids [142]. EVs from the endometrium carry different microRNAs and proteins, periodically regulated by the ovarian steroid hormones [143] (Table 2).

### 5.1. microRNA

Recent studies indicate that microRNAs establish immune tolerance at embryo implantation and pregnancy and may contribute to the regulatory effects of T regulatory and dendritic cells. MicroRNAs were also demonstrated to modulate inflammatory and hypoxic regulation in placentation [132]. During pregnancy, microRNAs contribute to endometrial receptivity, embryo implantation, placentation, following pregnancy, and labor course [144]. The microRNA-494, microRNA-923, and microRNA-30 families are expressed differentially in regulating endometrial receptivity through the leukemia inhibitory factor (LIF) [145]. Many studies show that particular microRNAs involve the modulation of the immune system during embryo implantation and pregnancy [146,147,148]. The importance of microRNAs in the regulation of immune responses through EVs during embryo implantation and pregnancy still needs to be investigated. In humans, a particular cargo of microRNAs in endometrial EVs has been demonstrated using the human endometrial cell line [149], while a total of 13 specific microRNAs were chosen and packaged in EVs. Bioinformatics analysis for microRNAs in the cargo of the EVs showed that EV-associated microRNAs regulate cell proliferation, inflammation, remodeling, and angiogenesis for embryo implantation. Some mammal studies showed that many microRNAs were recognized in the EVs from the uterine fluid of cyclic and pregnant mammals [150,151,152], and the EVs also contained some noncoding small nuclear RNA involved in the chemical modifications of other RNAs, indicating the effects of cellular function that may be regulated following the boosting of the EVs [149]. One microRNA can regulate many genes through modulating the transcription or translation. EVs usually carry many microRNAs; therefore, it is exciting to explore the physiological effects of the molecular cargo packaged within EVs for transport into the uterine microenvironment, which plays an important role in embryo implantation and pregnancy. Functional interactions between the uterus and embryo exist during embryo implantation and placenta development. EVs with microRNAs between human decidual stromal cells have been considered significant for embryo implantation and the programming of human pregnancy [7]. This study newly demonstrates that the miR-138-5p- and GPR124-adjusted NLRP3 inflammasome were identified in the EVs originated from human decidual stromal cells, suggesting that the miR-138-5p, GPR124, and NLRP3 inflammasome play a potential modulatory role in the decidual programming and placenta development of human pregnancy. These results reveal a new concept regarding the role of EVs, miR-138-5p, GPR124, and NLRP3 inflammasome in normal early pregnancy and spontaneous miscarriage. MicroRNA-138-5p acts as the transcriptional regulator of the G protein-coupled receptor 124 (GPR124). Evidence suggests EV-associated microRNA-138-5p regulates embryo implantation and early pregnancy by adjusting GPR124 and downstream signalings in human decidual stromal cells [7].

### 5.2. Protein 

In humans, proteomic analyses of endometrial EVs were demonstrated by mimicking the hormonal profile in the menstrual cycle through estrogen and progesterone treatment [153]. The proteomic analyses of endometrial EVs reveal 663 common proteins recognized in endometrial EVs, which contribute to EV biogenesis, trafficking, sorting, uptake, and recognition. In mammals, proteomic analyses of uterine EVs showed that 195 total proteins were identified [151], including macrophage migration inhibitory factor, peripheral plasma membrane protein CASK, apolipoprotein E, myosin light chain kinase, β actin, ATP citrate synthase, and glycogen phosphorylase. The differences between the EV-packaged proteins and EV function may result from the sensitivity of the proteomic profiling analysis, EV extraction, and enrichment methods. The effects of human endometrial cell-derived EVs on embryo implantation and pregnancy are likely modulated by the protein cargo of EVs, and packaged proteins can regulate embryo apposition, adhesion, and implantation through promoting extracellular matrix remodeling. EVs from steroid hormone-primed ECC-1 endometrial cells contain peptidases, collagens, integrins, galectins, and laminins. Meanwhile, trophectoderm markers of early implantation (cathepsin CTSC, actin ACTA2, and phosphoglucomutase 1[PGM1]) and implantation regulators (AHNAK, S100A10, and PLAT) are also identified through proteomic analysis [153,154]. Furthermore, in the human trophectodermal cell line, HTR8/SVneo cell-associated EVs were demonstrated to enhance the expression of N-cadherin in the primary decidual stromal cell model [139]. In summary, these studies demonstrate the functional role of EVs and their cargo as a mediator of embryo implantation and pregnancy through contribution to maternal–embryo communication.

### 5.3. Lipids

The cargo of EVs also contains lipids, which include lipids, lipid metabolites, and specific enzymes for lipid metabolism that can adjust the behavior and function of targeted cells [155]. There is evidence that RNA and protein are the main cargo of uterine EVs in embryo implantation and pregnancy. However, new evidence in mammals demonstrated that the lipid profiling of uterine EVs extracted from pregnant and cyclic mammals expressed eight classes of lipids, which vary significantly between EVs from pregnant and cyclic mammals, suggesting that these classes of lipids are involved in different pregnancy phases [156].

A few studies on the role of sphingolipids have demonstrated that sphingolipids isolated from pregnancy induce prostaglandin E2 secretion in decidual stromal cells [157], and sphingolipids are involved in decidual stromal cell apoptosis through inducing p38 MAPK phosphorylation and caspase 9 activations [158]. More studies showed that sphingolipids’ pathways and their receptors are individually modulated during the decidualization of human endometrial stromal cells [159], and the estrogen- and progesterone-primed decidualization of human endometrial stromal cells reveal different expressions of sphingolipids and their receptors during pregnancy progression in response to numerous pregnancy hormones. During embryo implantation and pregnancy, angiogenesis is a critical step to enhance decidualization and placentation. Some studies showed that sphingolipids regulate the endothelial cell invasion in human umbilical vein endothelial cells and ovine uterine artery endothelial cells [160], suggesting that sphingolipids induce angiogenesis for decidualization and placentation by functioning directly on the decidual stromal and endothelial cells. However, this still needs more studies to confirm.

## 6. Physiological and Immunological Functions of Extracellular Vesicles

In mammal studies, embryos and endometrium could release EVs into their surrounding environment, and these embryo- and endometrium-associated EVs possibly work together to affect embryo development, embryo–endometrial interaction, embryo implantation, and following pregnancy in a paracrine and/or autocrine manner.

### 6.1. The Immune Modulatory Effects of Extracellular Vesicles

In mammals, during embryo implantation and early pregnancy, the embryo locates and attaches to the endometrium, followed by the invasion of cytotrophoblasts to the decidua with communication with the maternal immune system. The maternal immune system usually needs to reply to embryo implantation timely and appropriately because of the direct communication between cytotrophoblasts and decidual immune cells [161,162,163]. Therefore, we focused on the immune modulatory effects of EVs during embryo implantation and early pregnancy, in which the maternal immune system tolerates the semiallogenic fetus and maintains the pregnancy through endometrial decidualization and placentation. The first study on maternal immune tolerance and modification to the semiallogenic fetus during pregnancy was reported in 1953 [164]. At the embryo–maternal junction, the maternal immune system is composed of many immune cells, which include monocytes/macrophages, natural killer cells, T lymphocytes, B cells, and dendritic cells [38]. A study identifies the effects of EVs in immune modulation during the gestational period through a proteomic analysis of protein cargo of EVs, which were isolated from the human placenta [134]. These proteins play an important role in the response to maternal immune tolerance and modulation during early pregnancy. Some studies reported the immune-suppressive effect of placenta-isolated EVs, which contain TNF-related apoptosis-inducing ligands (TRAIL) and Fas ligands (FasL). These proteins provide the local immune tolerance at the fetal–maternal junction by inducing the apoptosis of T cells [165]. HLA-G is released from the EVs, which subsequently declines as the pregnancy progresses in human pregnancies [166]. Some studies revealed that placenta-derived EVs are associated with the Treg cell differentiation and regulation process, thus involving immune tolerance [167]. It has been demonstrated that the stable release of embryo and endometrial EVs offers a protective environment for maternal–fetal communication during embryo implantation and pregnancy. It is well known that immunological tolerance to the fetal allograft must provide the development for conceptus and the subsequent maintenance of pregnancy. During the embryo attachment period, EVs seem to transport molecules potentially able to modulate the local endometrial immune system [168,169,170,171]. This stimulated or inhibited modulation appears to depend on the cargo of EVs, such as microRNAs or proteins.

There are also data supporting the notion that the trophoblasts secrete many molecules that govern the polarization of the decidual macrophage M1/M2 phenotype to modulate the maternal immune tolerance to the fetus [172,173]. EVs seem to control the crosscommunication between the trophoblasts and the immune cells. Through uptaking trophoblast-derived EVs, the monocytes seem to increase the migration and production of cytokines [174]. The progesterone-induced blocking factor (PIBF) has been considered to control NK cell activity. PIBF is identified in embryo-derived EVs, which adhere to the surface of CD4+ and CD8+ peripheral T cells and induce IL-10 synthesis [175]. This function of embryonic EVs can be inhibited by preincubation of EVs with an antiPIBF antibody, suggesting the communication between the embryo and the maternal immune system through EVs during early implantation and pregnancy [175]. Therefore, these observations imply that embryo- and endometrial-derived EVs can recruit and regulate monocytes in a cell contact-independent manner.

### 6.2. Communication between the Embryo and the Maternal Immune System through Extracellular Vesicles

In mammals, the preimplantation embryos stay in the uterine cavity for a while, and EVs may play a critical role in embryo–endometrial communication for implantation (Figure 3). The human ECC1 cell model demonstrates that endometrial EVs modulate the trophectodermal function, such as adhesive ability [153]. Meanwhile, through proteomic analysis of these endometrial EVs, many important proteins were identified, including fibronectin, total FAK, and phosphorylated FAK proteins, suggesting that these proteins likely contribute to promoting the adhesive capacity of the trophoblasts following EV uptake [153]. Furthermore, the FAK pathway has been demonstrated to play a crucial role in EV interaction within the blastocyst intercellular communication between the trophectodermal and the inner-cell-mass cells [176]. In mammal studies, endometrial EVs function as the embryo–endometrial interaction and enhance endometrial receptivity during embryo implantation and pregnancy [139,154,177,178,179]. Pretreatment with estradiol and progesterone in immortalized endometrial cell lines can alter the molecular features of EV-modulated endometrial receptivity [154,179,180]. During embryo implantation, an increased expression of CD63 and higher numbers of EVs in human endometrium were demonstrated [150]. Additionally, the functional effects of EVs from human primary decidual stromal cells on the trophectoderm that are consistent with the impact on embryo implantation and pregnancy have been demonstrated [139]. Evidence by using immortalized trophectodermal cell lines indicates that the cargo of EVs could potentially serve as regulators of endometrial receptivity for embryo implantation after EV treatment and may provide the concept in the differential diagnosis of female infertility [181]. Consequently, this recent evidence implies the potential role of EVs as modulators of embryo implantation through the contribution to embryo–endometrium interaction, indicating that the analysis of EV cargo could offer a new concept of endometrial receptivity assay in the infertility and reproductive field [182].

## 7. Conclusions 

In EV research in reproductive biology, it is well-documented that proper biochemical and cellular interactions between the embryos and the uterine endometrium are required for embryo implantation and pregnancy. Recent studies suggest that endometrial EVs could have autocrine and/or paracrine physiological functions at different periods of pregnancy. There is considerable evidence that endometrial EVs are correlated and synchronized with ovarian steroid hormones and trophectoderm-originated EV cargo, including microRNAs, proteins, and lipids in the physiological microenvironment of embryo–endometrial communication and maternal immunomodulation for embryo implantation and subsequent pregnancy. Although the mechanisms of the actions in biogenesis and cargo of endometrium- or trophoblast-derived EVs still need to be established, new technological approaches in EVs will shed light on their functions and underlying mechanisms in the processes of embryo implantation and pregnancy. Furthermore, the exploration of EVs could provide a practical tool for diagnostic or therapeutic approaches in reproductive medicine and infertility. Further research is required to identify the differences in the cargo of EVs between physiological and pathological embryo–maternal interactions, which investigate the factors of infertility and improve embryo implantation and subsequent pregnancy outcomes.

## Figures and Tables

**Figure 1 ijms-23-13382-f001:**
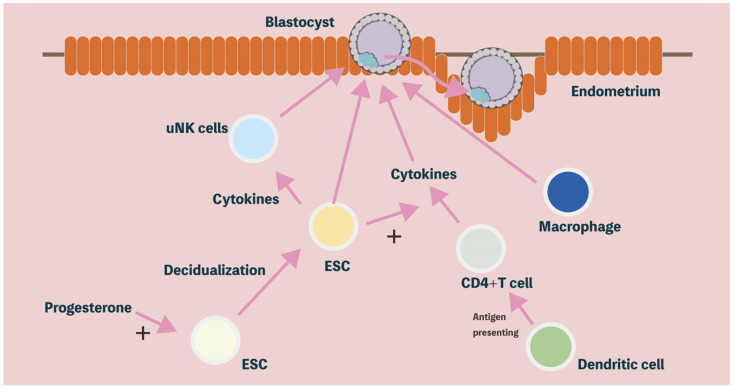
Endometrial receptivity: decidualization, immune cells, cytokines, and immune modulatory properties determine reproductive outcomes. Under the influence of progesterone, ESCs complete decidualization. The decidualized ESCs are critical to the development of surrounding trophoblastic, hematopoietic (e.g., uNK and T cell), and mesenchymal cells (e.g., blood vessels), which play an important role in blastocyst implantation. ESCs: endometrial stromal cells; uNK: uterine natural killer.

**Figure 2 ijms-23-13382-f002:**
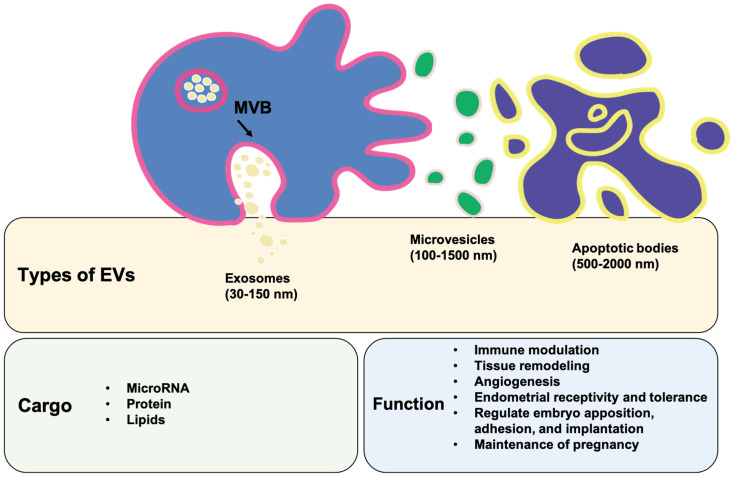
Characteristics, cargos, and functions of extracellular vesicles. The three main subtypes of extracellular vesicles (EVs) are exosomes, microvesicles (MVs), and apoptotic bodies, which are generally differentiated through size, content, and function examination. MVB: multivesicular body.

**Figure 3 ijms-23-13382-f003:**
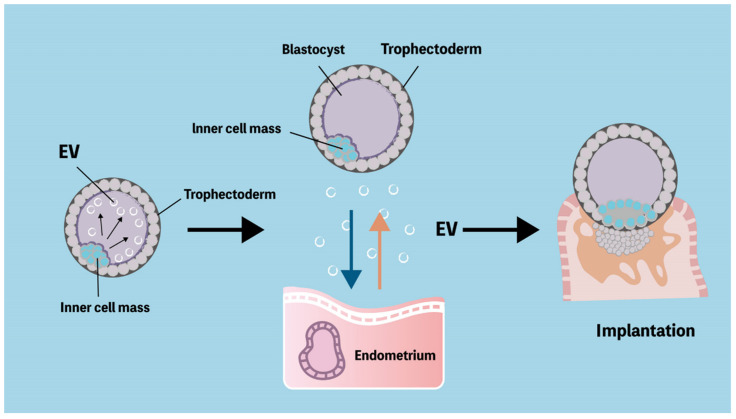
Extracellular vesicles provide intercellular communication both between embryos and endometrium. Extracellular vesicles (EVs) extend across the blastocoel and link the mural trophectoderm and inner cell mass. Trophectoderm cell line releases miRNA-containing EVs that stimulate endometrial epithelium. The communication between blastocyst and endometrium triggers successful embryo implantation.

**Table 2 ijms-23-13382-t002:** Characteristics and cargo of extracellular vesicles.

	Characteristics			
Contents	Origin	Size of EV	Member	Role
microRNA [132,144,145,146,147,148,149,150,151,152]	exosomes, microvesicles	30–1500 nm	mcroRNA-494, microRNA-923, microRNA-30 family, microRNA-138-5p…	immune regulation, endometrial receptivity regulation
Protein [151,153,154]	exosomes, microvesicles	30–1500 nm	663 common proteins	regulate embryo apposition, adhesion, and implantation
Lipids [155,156,157,158,159,160]	exosomes, microvesicles	30–1500 nm	eight classes	angiogenesis, decidual stromal cell apoptosis

Note: EV, extracellular vesicle.

## Data Availability

Not applicable.

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
