# Peer review of "Immune Tolerance of Embryo Implantation and Pregnancy: The Role of Human Decidual Stromal Cell- and Embryonic-Derived Extracellular Vesicles"

_ijms, 2022, doi:10.3390/ijms232113382_

Round 1

Reviewer 1 Report

Journal: International Journal of Molecular Sciences

Manuscript ID: ijms-1916988

Manuscript title: Immune tolerance of embryo implantation and pregnancy: The

role of human decidual stromal cells- and embryonic-derived extracellular vesicles.

Authors: Wu H-M, et al.,

Article type: Review

General comments:

The review intend to focus on immune tolerance of embryo implantation and pregnancy and particularly the role of embryonic-derived extracellular vesicles.

The authors indicated “In this review, we will highlight the roles of human decidual stromal cells-associated EVs in the immune tolerance of embryo implantation and pregnancy”. But this review fail to do so.

Poorly written. Language correction is needed. The review is vaguely focusing on immune modulation. The flow of the review is distressing.

Fetal allograft and rejection mechanism is one of the most important component but is not discussed in detail in this review.

Table 1 and 2 must include references.

The figures should have footnote explaining what the figure depicts for the  readers to understand it better.

Recent references on EV and fetal rejection were not included

Some references that may be useful.

Ober C. (1998). HLA and pregnancy: the paradox of the fetal allograft. American journal of human genetics62(1), 1–5. https://doi.org/10.1086/301692

Erlebacher A. (2010). Immune surveillance of the maternal/fetal interface: controversies and implications. Trends in endocrinology and metabolism: TEM21(7), 428–434. https://doi.org/10.1016/j.tem.2010.02.003

Mincheva-Nilsson L. (2006). Immune cells and molecules in pregnancy: friends or foes to the fetus?. Expert review of clinical immunology2(3), 457–470. https://doi.org/10.1586/1744666X.2.3.457

Benichou, G., Wang, M., Ahrens, K., & Madsen, J. C. (2020). Extracellular vesicles in allograft rejection and tolerance. Cellular immunology349, 104063. https://doi.org/10.1016/j.cellimm.2020.104063

Ou, Q., Dou, X., Tang, J., Wu, P., & Pan, D. (2022). Small extracellular vesicles derived from PD-L1-modified mesenchymal stem cell promote Tregs differentiation and prolong allograft survival. Cell and tissue research389(3), 465–481. https://doi.org/10.1007/s00441-022-03650-9

Cho, K., Kook, H., Kang, S., & Lee, J. (2020). Study of immune-tolerized cell lines and extracellular vesicles inductive environment promoting continuous expression and secretion of HLA-G from semi-allograft immune tolerance during pregnancy. Journal of extracellular vesicles9(1), 1795364. https://doi.org/10.1080/20013078.2020.1795364

Specific comments:

Abstract:

Line 12 to 17: Redundant.

Line 13: and establishment of successful pregnancy

Line 13 and 14: Change to “Successful pregnancy outcomes involves maternal immune modulation during embryo implantation”.

Line 19 to 21: Rephrase.

Line 24 to 27: Redundant.

Introduction:

Line 33: Remove “subsequent”

Line 33: “Maternal immune modulation” be specific general? or local??

Line 33 to 35: Rephrase.

Line 35 & 36:

Rephrase. “Since the first human IVF baby was born in 1978, pregnancy and live birth rates following assisted reproductive technologies (ART) has improved significantly”.

Line 36 to 38. The sentence seems incomplete.

Clarify how Section 2 is important and  plays a key role in immune modulation. Instead of repeating what is in the literature.

Thea authors could have done better job in Section 3. This section is foundation of this review. Key elements of immune modulation is lacking in this section.

Title of table 1 need to be modified. “Immune modulatory properties” not well explained.

Line 196-197

What Th1 and Th2 cytokine play major role? Please clarify and list them. Why some cytokines are increased, and some others are decreased during implantation? How they play crucial role in maternal-fetal immune tolerance

Some discussions are mere mentioning of cytokines, cells and genes rather than how immune modulation mediates these important steps for eg. trophoblast invasion

Author Response

Please see the attachment, thanks.

Reviewer 2 Report

The work presented by Wu and coworkers summarizes in an organized way and with all kinds of specifications, the role of human decidual stromal cells-associated EVs in the immune tolerance of embryo implantation and pregnancy.
Some comments:
-Lines 59, 117, 355 and 523: reference to figures and tables cannot be in section title, but in the body text.
- Figure 2: the letters in the box "funtion" are too small, and this part of the figure is hard to read.
- Please provide a brief summary of challenges/gaps in the field and
suggest the way ahead with respect to future directions with this work.
-The authors should add more figures to make the manuscript more interesting and attractive for potential readers.
- Abbreviations are not appropriately defined throughout the
manuscript. I reccoment to prepare the list of all abreviation and include it in the revised version of manuscript.
- The Authors should run spell check and carefully check for typos.
- If the figures are taken from published sources should mention “reprinted after permission from authors” statement as part of the figure legend or if they are prepared based on other papers, the reference is required.
